# Combination treatment to improve mucociliary transport of *Pseudomonas aeruginosa* biofilms

**Kaitlyn R. Rouillard**[1], **Christopher P. Esther**[1], **William J. Kissner**[1], **Lucas M. Plott**[1],
**Dean W. Bowman**[1], **Matthew R. Markovetz**[1], **David B. Hill**[1,2]*

**1** Marsico Lung Institute, UNC Chapel Hill, Chapel Hill, NC, United States of America, **2** Joint Department of Biomedical Engineering, UNC Chapel Hill, Chapel Hill, NC, United States of America

* dbhill@med.unc.edu

**Data Availability Statement:** Data is available on the Carolina Digital Repository at https://doi.org/10.17615/kx69-5d74.

## Abstract

People with muco-obstructive pulmonary diseases such as cystic fibrosis (CF) and chronic obstructive pulmonary disease (COPD) often have acute or chronic respiratory infections that are difficult to treat due in part to the accumulation of hyperconcentrated mucus within the airway. Mucus accumulation and obstruction promote chronic inflammation and infection and reduce therapeutic efficacy. Bacterial aggregates in the form of biofilms exhibit increased resistance to mechanical stressors from the immune response (e.g., phagocytosis) and chemical treatments including antibiotics. Herein, combination treatments designed to disrupt the mechanical properties of biofilms and potentiate antibiotic efficacy are investigated against mucus-grown *Pseudomonas aeruginosa* biofilms and optimized to 1) alter biofilm viscoelastic properties, 2) increase mucociliary transport rates, and 3) reduce bacterial viability. A disulfide bond reducing agent (tris(2-carboxyethyl)phosphine, TCEP), a surfactant (NP40), a biopolymer (hyaluronic acid, HA), a DNA degradation enzyme (DNase), and an antibiotic (tobramycin) are tested in various combinations to maximize biofilm disruption. The viscoelastic properties of biofilms are quantified with particle tracking microrheology and transport rates are quantified in a mucociliary transport device comprised of fully differentiated primary human bronchial epithelial cells. The combination of the NP40 with hyaluronic acid and tobramycin was the most effective at increasing mucociliary transport rates, decreasing the viscoelastic properties of mucus, and reducing bacterial viability. Multimechanistic targeting of biofilm infections may ultimately result in improved clinical outcomes, and the results of this study may be translated into future in vivo infection models.

## Introduction

Muco-obstructive pulmonary diseases (MOPD) are characterized by the accumulation of hyperconcentrated mucus within the airway that promotes chronic inflammation and facilitates infection by opportunistic pathogens including *Pseudomonas aeruginosa* [1–3]. The pathological biophysical properties of mucus are primarily dependent on the total % solids concentration which ranges from ~2% to upwards of 8% in severe disease [1, 4, 5]. Secreted

**Funding:** The authors report the following sources of funding: Cystic Fibrosis Foundation, (ROUILL22F0) awarded to KRR, Cystic Fibrosis Foundation (HILL19G0) awarded to BDH, Cystic Fibrosis Foundation (HILL20Y2-OUT) awarded to BDH, Cystic Fibrosis Foundation (BOUCHE19R0) awarded to BDH, National Heart, Lung, and Blood Institute (1P01HL164320) awarded to RCB, and National Institute of Diabetes and Digestive and Kidney Diseases (P30DK065988) awarded to RCB.

**Competing interests:** No authors have competing interest.

mucin glycoproteins are the primary high molecular weight, polymeric component of mucus that give it its characteristic biophysical properties which dictate mucociliary clearance [5, 6]. Airway mucus is composed of a mucin network of MUC5AC and MUC5B mucins that are secreted in response to chemical and mechanical stressors (such as inhaled pathogens and environmental stimuli) and as part of maintaining homeostasis, respectively [6–8]. In MOPD, the concentrations of both mucins increase and significantly impact the biophysical properties of mucus [4, 5]. Greater mucus viscoelastic moduli are associated with poor mucociliary transport (MCT) [4, 9] and ultimately declining lung function [10].

Compromised MCC allows for colonization by inhaled pathogens, which aggregate and produce an exterior matrix of polymers including lipids, polysaccharides, proteins, and DNA [11–14]. Previously, single cell transcriptomics analysis of *Pseudomonas aeruginosa* biofilm populations has quantified an upregulation of matrix components such as polysaccharides and multi-drug efflux genes [15], allowing for aggregates to be protected from the immune response and antibiotic treatment. Biofilms are too large (>5 μm) and mechanically robust to be phagocytosed and may persist in the airway in perpetuity [16–19]. Contributing to biofilm persistence is the failure of most conventional antibiotics to alter the mechanical properties of biofilms despite biocidal action and significant cell death [20–22]. Reduced penetration through and efficacy in airway mucus also limits therapeutic utility of conventional antibiotics [23, 24]. For example, tobramycin exhibits reduced antibiotic action in sputum and does not disrupt the viscoelastic properties of *P. aeruginosa* biofilms grown in airway mucus [21, 24, 25]. It has been proposed that disruption of the biofilm matrix may potentiate antibiotic efficacy and improve the potential for clearance [25, 26], which has previously been observed with the use of alginate oligosaccharides [27–31]. Enzymatic degradation of DNA in the matrix via DNase has also demonstrated synergistic bactericidal action when used in combination with antibiotics [26]. However, other work has shown that *P. aeruginosa* biofilms grown in disease-mimicking mucus (5% solids) were resistant to degradation via DNase and showed no decrease in viscoelastic moduli at clinically relevant doses [32].

Disruption of *P. aeruginosa* biofilms via poly-L-lysine or hyaluronic acid potentiated antibiotic action by tobramycin and altered biofilm complex viscosity ($\eta^*$) despite having no significant effect on bacterial viability when dosed alone [25]. Concomitant delivery was also associated with increased mean MCT rates, though transport was largely heterogeneous. The manipulation of mucus and biofilm mechanics for improving antibiotic efficacy has been investigated through a broad range of mechanisms including treatment with a surfactant [33, 34], disulfide bond reduction [25, 32, 35], enzymatic degradation of matrix components [26, 36, 37], and biopolymer intercalation/matrix destabilization [25, 29]. Ultimately, a combination of these mechanisms may be necessary for complete eradication of biofilms from the airway in MOPD.

To assess the disruptive capacity of combination treatment, biofilm rheology and transport were characterized using in vitro models. Mucociliary transport in human bronchial epithelial (HBE) cultures can be efficiently measured in the recently developed "racetrack" mucociliary transport device (MCTD) [9, 25]. Coordinated ciliary beating around the circumference of the culture allows for quantification of mucus and biofilm transport rates. *Pseudomonas aeruginosa* biofilms were grown in HBE mucus at 3% solids, representative of mild disease, and transport in the MCTD was quantified as a function of treatment conditions with a surfactant (NP40), a reducing agent (tris(2-carboxyethyl)phosphine, TCEP), a DNA degradation enzyme (DNase), a biopolymer (hyaluronic acid, HA), and/or an antibiotic (tobramycin). The evaluation of biofilm MCT using the MCTD represents a valuable screening tool for potential therapeutics for treating respiratory infections in MOPD. Effective dual and triple combination treatments may be translated to murine models in future work to reduce mucus viscosity, increase mucociliary transport, and clear respiratory infection.

## Methods

### Materials

Unless otherwise specified, all materials were used as received. The strain PAO1 of *P. aeruginosa* was obtained from the ATCC. Non-diseased human bronchial epithelial cells were purchased from the Marsico Lung Institute Tissue Procurement Core and cultured at the air liquid interface as previously described [4, 5]. The UNC IRB determined that the use of these cells constituted exempt research (23–0186). The surfactant nonyl phenoxypolyethoxylethanol (NP40) was a kind gift from Dr. Brian Button at UNC Chapel Hill. The reducing agent TCEP, the DNA degradation enzyme DNase, carboxylated 1 μm polystyrene microspheres, and common laboratory chemicals were purchased from MilliPore Sigma [25, 32]. Hyaluronic acid (ultra-low molecular weight) with a molecular weight of <6 kDa was purchased from LotionCrafters and its characterization has been previously described [38].

### Ciliary beat frequency and mucociliary transport quality control

Quantification of ciliary beat frequency (CBF) was performed using 40x objective of a Nikon TE2000U microscope under brightfield illumination. Fluorescent 1 μm microspheres were diluted 1:1200 from stock into PBS and 200 μL was added to the racetrack. Transport of microspheres in PBS was recorded using the 10x objective as previously described [9, 25]. Three videos from two separate quadrants of each racetrack were recorded to ensure quality control within and between cultures. Values of CBF and MCT were quantified using custom MATLAB (© 2022 The Math-Works, Natick, MA) scripts which are available upon request. Stuck beads were defined as those having MCT rates less than 0.005 mm/min and were removed from the analysis. Statistical significance was performed with single factor ANOVA with post hoc Tukey analysis.

### Biofilm growth and treatment

An overnight culture of PAO1 was cultured in tryptic soy broth (TSB) from frozen stock at 37˚C, resuspended in fresh TSB, and cultured to an $OD_{600}$ of 0.25, indicative of $10^8$ CFU/mL [21]. Bacteria were diluted 100-fold into 3% HBE mucus, which was prepared as previously described and characterized with standard quality assurance and quality control measures [4, 5]. The bacteria-mucus mixture was aliquoted (60 μL) into the center wells of a 96-well plate with 1 μL of yellow-green fluorescent carboxylated 1 μm polystyrene microspheres (diluted 1:10 from stock) and surrounding wells were filled with sterile deionized (DI) water to prevent evaporation [25, 32]. The well plate was incubated with shaking (100 rpm) for 24h at 37˚C to form a viscous biofilm aggregate that was optically turbid and easily separated from remaining growth media using a positive pressure pipette (50 μL). PAO1 biofilms were treated with single agents or dual or triple combinations according to Table 1. NP40 was dosed at 0.1%, which is below the critical micelle concentration, to avoid any contributions from micelle formation. TCEP was dosed at 10 mM, HA at 1 mg/mL, and DNase at 1 mg/mL, which were previously found to be effective in PAO1 biofilms grown in 4% HBE [25]. Treated biofilms were incubated with shaking (100 rpm) at 37˚C for 24h. Biofilms were diluted 10x in sterile DI water, disrupted with trituration and vortexing, serially diluted in DI water, and plated on tryptic soy agar (TSA) to quantify biofilm viability and qualitatively assess colony morphology and screen for contamination.

### Particle tracking microrheology

Fluorescence video microscopy was used as previously described to quantify the viscoelastic properties of PAO1 biofilms grown in mucus via particle tracking microrheology (PTMR) [4,

**Table 1. Matrix of biofilm treatment combinations.**

| Treatment | Surfactant | Reducing Agent | Biopolymer | Enzyme | Antibiotic |
|---|---|---|---|---|---|
| Single Agent | NP40 0.1% | TCEP 10 mM | HA 1 mg/mL | DNase 1 mg/mL | Tobramycin |
| Dual | NP40+Tob | TCEP+Tob | HA+Tob | DNase+Tob | |
| Triple | HA+NP40+Tob | | | | 1 mg/mL |
| | TCEP+ NP40+Tob | | | | |
| | | TCEP+HA+Tob | | | |

25, 39]. Briefly, a portion of the biofilm (5 μL) was added to a glass slide inside a parafilm window and sealed with a coverslip. The Brownian diffusion of incorporated microspheres was recorded with videos and tracked and quantified with custom Python and MATLAB scripts, which are available upon request. The mean squared displacement of each particle was used to calculate viscoelastic moduli as previously described [25, 39]. Statistical significance was determined using single factor ANOVA with post hoc Tukey analysis on the ensemble $\eta^*$ values determined from each slide (n≥6).

## Cone and plate rheometry

Macroscopic rheology was characterized with a 20 mm 1˚ cone and plate rheometer (DHR-3, TA Instruments) as previously described [4, 25, 32]. The linear viscoelastic regime was determined via strain sweeps at 0.5 Hz and 5 Hz and subsequent frequency sweeps were performed at strains where viscoelastic moduli behaved linearly. Quantification of viscoelastic moduli was performed at 1 Hz, representative of tidal breath.

## Mucociliary transport

Mucociliary transport device (MCTD) "racetrack" cultures were maintained at the air liquid interface (ALI) and washed weekly [9]. Before each analysis, the cultures were washed thrice with short (30s) washes and one long (60 min) wash with PBS to remove endogenous mucus. The biofilm (20 μL) was added to quadrant 1 (Q1) and allowed to equilibrate in the incubator for 10 min prior to microscopy. Using a 10x objective, six videos of the biofilm were recorded from Q1 and Q2 (counterclockwise, Fig 1). To facilitate higher throughput analysis, a second biofilm was added to Q3 at the same time as the first biofilm and videos were recorded from Q3 and Q4. Biofilms were evaluated in biological duplicate from two separately prepared and treated batches of biofilms (n = 4) in separate MCTD. Rates of MCT were quantified using custom Python and MATLAB scripts as previously described [9, 25] which are available upon request. Stuck beads were defined as those having MCT rates <0.1 μm/s and were removed from analysis. Of note, more viscoelastic materials were associated with greater numbers of stuck beads while disrupted biofilms (e.g., treated with triple combinations) exhibited lower complex viscosity values and fewer stuck beads. The number of filtered beads from treated biofilms is included in S1 Table in S1 File. Statistical significance was determined with single factor ANOVA with post hoc Tukey-Kramer analysis.

# Results

## Cilia beat frequency and mucociliary transport are consistent across racetrack quadrants and cultures

The MCTD allows for greater coordinated cilia beating than conventional HBE cultures and facile quantification of MCT rates (Fig 2). Cilia beat frequency was determined via a

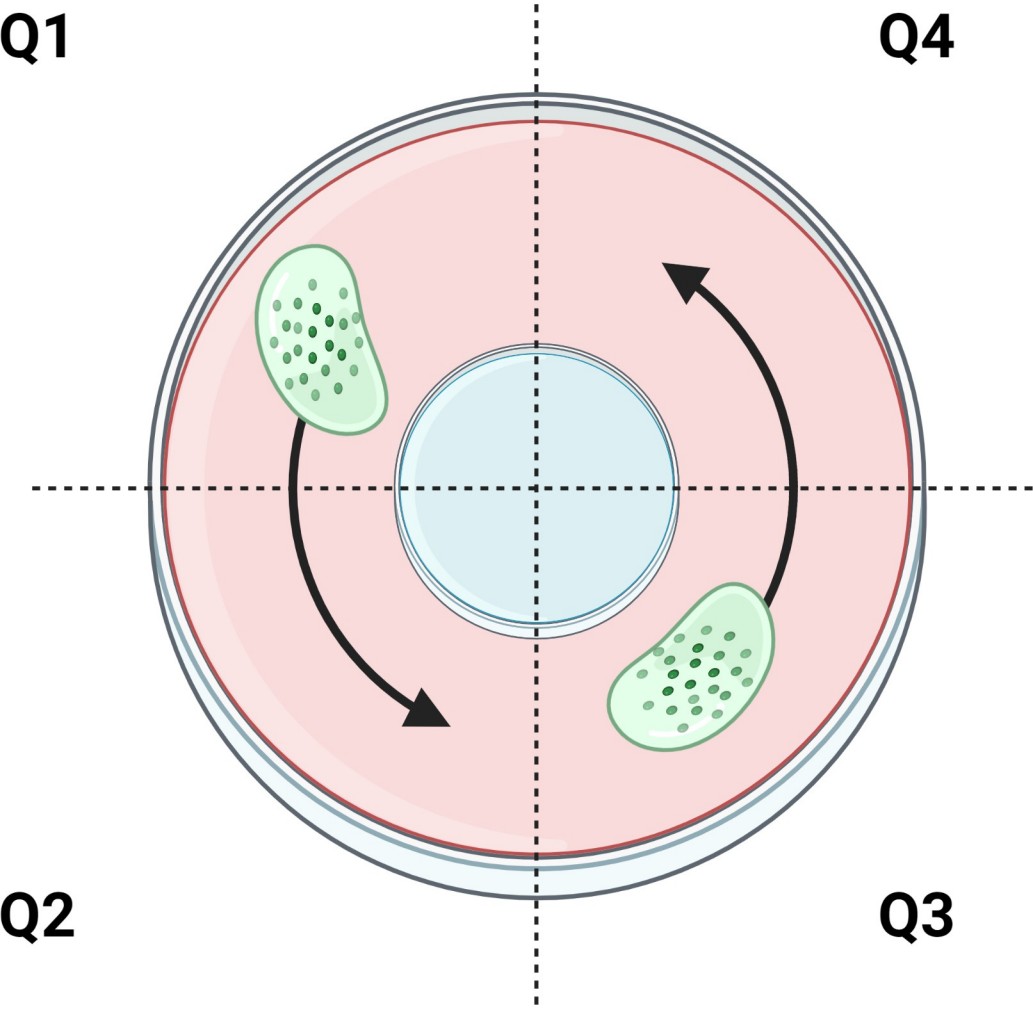

**Fig 1. Representation of MCTD imaging for transport of biofilms.** MCT occurs counterclockwise with n≥3 videos per quadrant per biofilm. Figure created with biorender.com.

periodogram wherein CBF was defined as the frequency peak closest to 0 Hz (after the initial drop from noise) with the greatest signal intensity (S1 Fig in S1 File). CBF was consistent across quadrants and cultures (Fig 2A and 2B). Rates of MCT were more heterogeneous than CBF (Fig 2C), though MCT rates were not significantly different between cultures. When comparing the transport of beads in PBS across five cultures, there was no significant difference in MCT rates after filtering out the stuck beads (i.e., MCT <0.1 μm/s).

## Mucus viscoelasticity and mucociliary transport rates are inversely related

With increasing mucus concentration, viscoelastic moduli increased (Fig 3A), consistent with previous work [4, 5, 9, 39]. Treatment of mucus with TCEP (10 mM) caused a significant (p<0.01) reductions in mucus $\eta^*$ (Fig 3B) for all tested concentrations of mucus. There was no significant difference in MCT rates for mucus as a function of concentration but reduction of disulfide bonds with TCEP treatment significantly increased MCT rates (Fig 3C). Analysis of mucus $\eta^*$ and MCT rates indicates a power-law relationship between mucus complex viscosity and transport rates in the MCTD (Fig 3D). These data indicate that for HBE mucus, $\eta^*$ may be

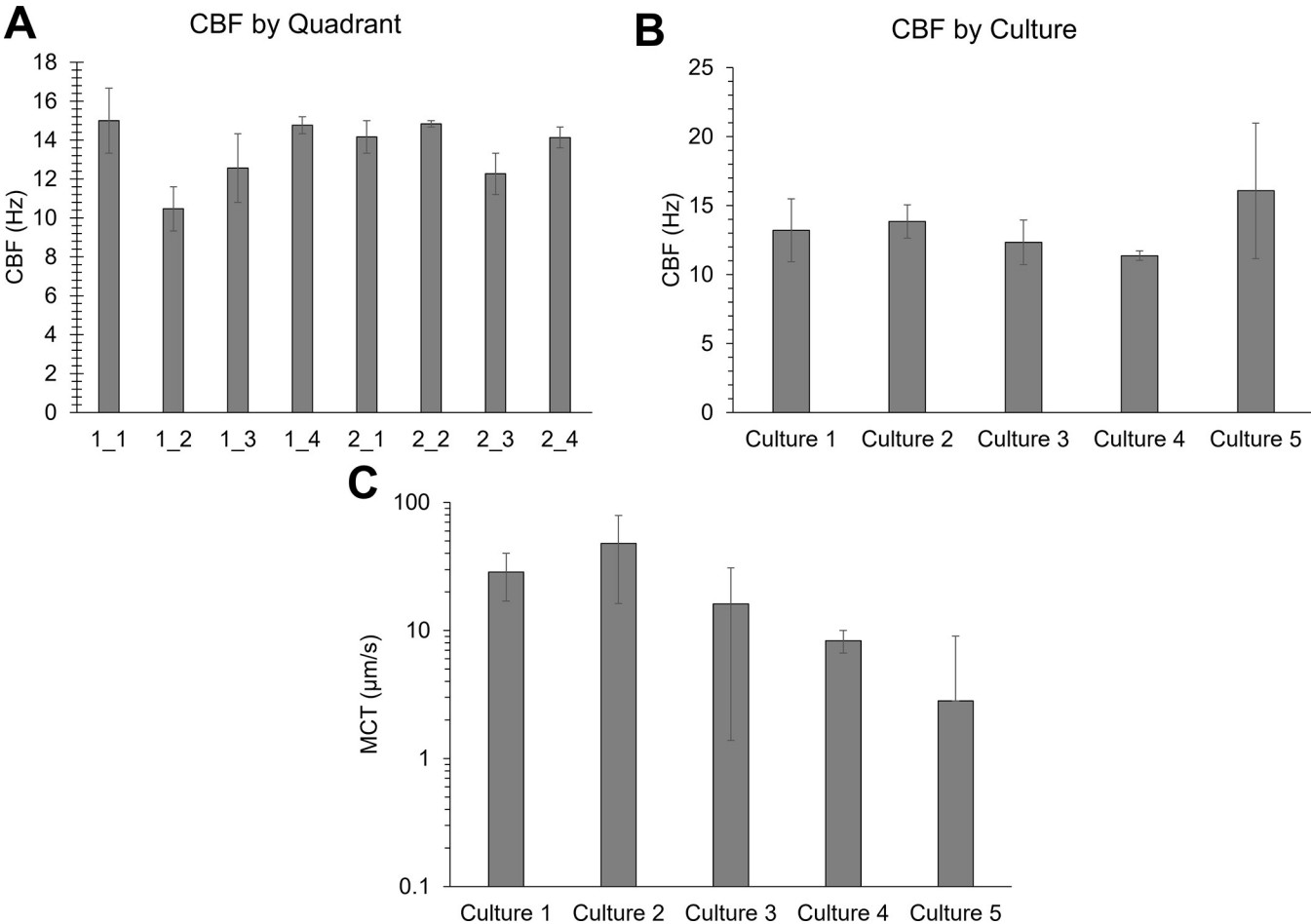

**Fig 2. Characterization of cilia beat frequency and mucociliary transport in racetrack cultures.** A) Average cilia beat frequency for each quadrant of two separate cultures. B) Overall average cilia beat frequency from five cultures. C) Mean MCT rates of beads in PBS for five separately prepared and evaluated cultures. Data is representative of the average and standard deviation of $\geq$3 individual measurements.

used to approximate MCT rates using a power-law relationship: MCT = $0.759(\eta^*)^{-0.699}$ (Fig 3D) with an $R^2$ value of ~0.87.

## Biofilm disruption and reduced viability do not necessarily result in decreasing viscoelasticity

Biofilm growth in mucus was associated with biophysical properties that were distinct from uninfected mucus and mucus with planktonic bacteria (Fig 4). Biofilms grown in 3% mucus (PAO1 3%) had a greater mean $\eta^*$ compared to 3% mucus (3% HBE) but lower than that of 3% mucus with planktonic bacteria (3%+PAO1, Fig 4A). Additionally, the presence of a watery component was detected via Gaussian mixture modeling analysis of the biofilm (Fig 4B) that was not present in 3% HBE with or without planktonic bacteria. It is hypothesized that this watery component probed by PTMR is from water-filled regions of the biofilm that have previously been shown to serve as reservoirs and transport channels [40]. In agreement with previous work [20–22, 25], changes in biofilm mechanical properties due to treatment (Fig 4C) were not necessarily indicative of decreased bacterial viability (Fig 4D) and vice versa. Treatment with NP40, TCEP, or DNase was associated with overall (but not significant) decreased

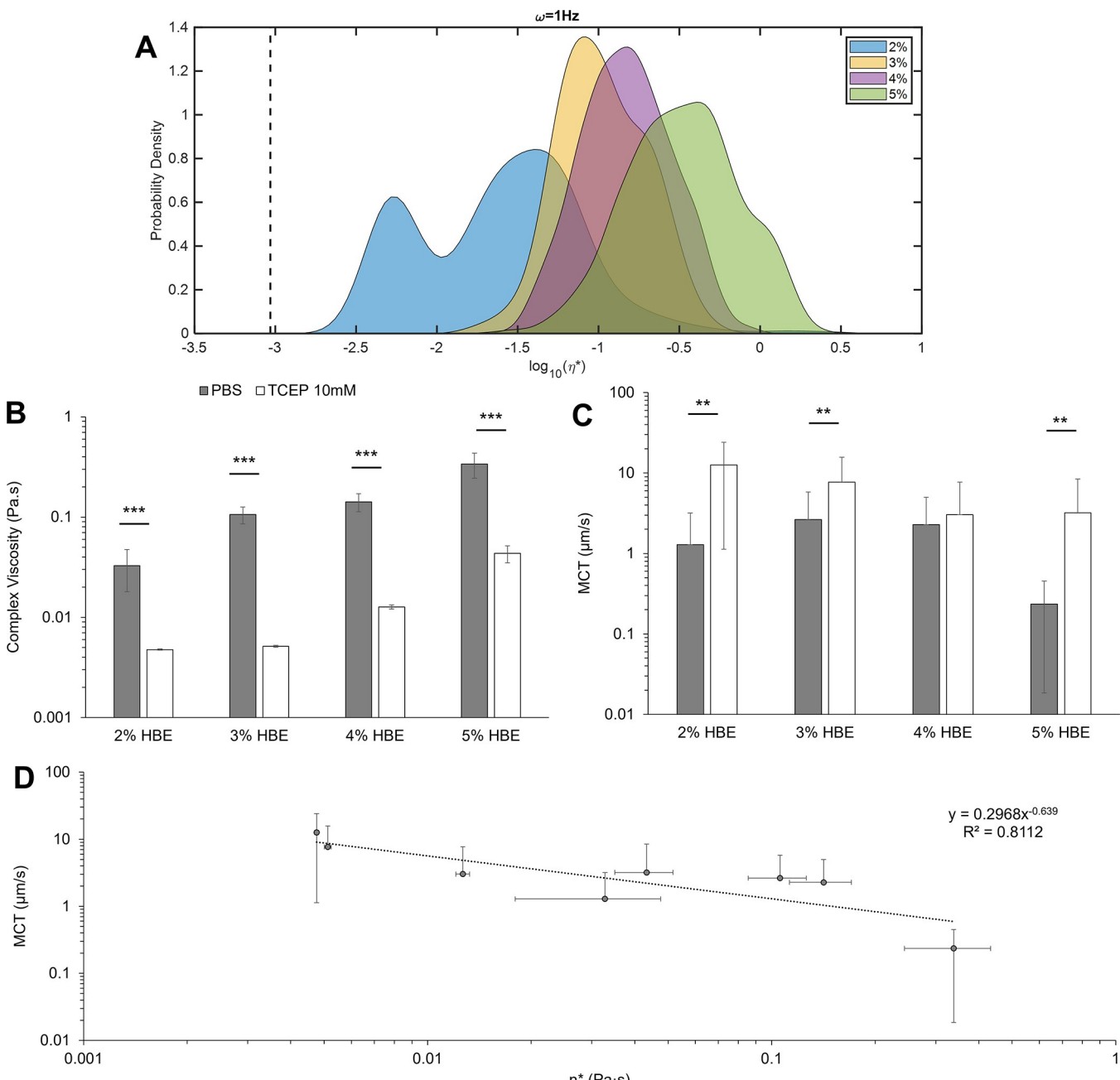

**Fig 3. HBE mucus rheology and transport as a function of concentration and treatment conditions.** A) Mucus complex viscosity distribution with % solids concentration. Data is representative of the distribution of every tracked particle complex viscosity. B) Mucus ensemble complex viscosity as a function of % solids ± treatment with 10 mM TCEP. Data is presented as the mean ± standard deviation of complex viscosity values for n≥3 separately prepared and evaluated mucus samples. Statistical significance was determined using single factor ANOVA with post hoc Tukey analysis. C) MCT rates of mucus as a function of % solids ± treatment with 10 mM TCEP. Data is presented as the mean ± standard deviation of MCT rates for n≥6 videos of mucus transport. Statistical significance was determined using single factor ANOVA with post hoc Tukey Kramer analysis. D) Power law relationship between mucus complex viscosity values and measured MCT rates in racetrack cultures.

$\eta^*$ while HA treatment resulted in significantly increased $\eta^*$. Further divergence from any noticeable trend was observed in combination treatment with tobramycin. No significant change in biofilm $\eta^*$ was measured after dual treatments, and only HA+Tob resulted in a significant change from single agent treatment alone. The heterogeneity in biofilm viscoelastic

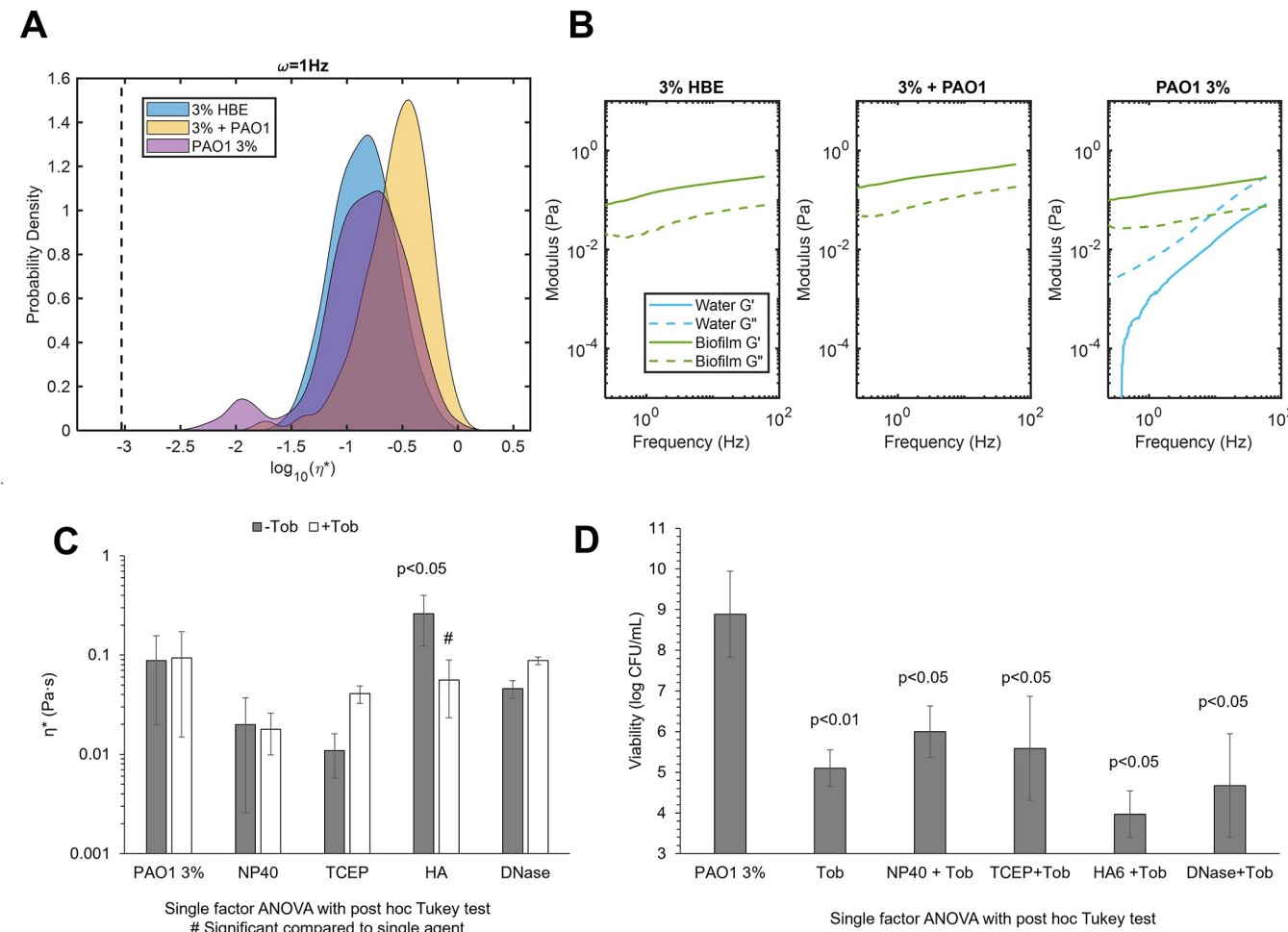

**Fig 4. Biofilm rheology and viability as a function of treatment conditions.** A) Complex viscosity of 3% HBE, 3% HBE + PAO1 planktonic, or PAO1 3% biofilms. Data is representative of every tracked bead for three separately prepared and evaluated specimens. B) Elastic (solid) and viscous (dashed) moduli of mucus alone, mucus with planktonic PAO1, or PAO1 biofilm grown in 3% mucus. Data is representative of mean moduli for every tracked bead for three separately prepared and evaluated specimens. C) Biofilm mean complex viscosity as a function of matrix disruption ± tobramycin. Statistically significant differences between untreated and treated biofilms were calculated using the single factor ANOVA with post hoc Tukey analysis and # indicates significant differences from the single agent treatment. Data is presented as the mean ± standard deviation of complex viscosity values for n≥3 separately prepared and evaluated biofilms. D) PAO1 biofilm viability as a function of matrix disruption ± tobramycin. Data is presented as the mean ± standard deviation of viability for n≥3 separately prepared and evaluated biofilms. Statistical significance was determined via single factor ANOVA with post hoc Tukey analysis.

properties after treatment has previously been demonstrated in mucus-grown *P. aeruginosa* biofilms [25, 32] and is visualized in S2 Fig in S1 File which shows every tracked particle η* for each treatment condition. Notably, the prevalence of the watery component changes with treatment condition with no discernable pattern. Viability ranged from $10^4$ to $10^6$ CFU/mL after combination treatment with the greatest reduction achieved with HA+Tob (Fig 4D). Together these data demonstrate that mechanical properties and bacterial viability are not necessarily linked.

To further compare biofilm mechanical properties after treatment, the non-Gaussian parameter κ was quantified to describe heterogeneity wherein values >>1 are increasingly heterogeneous [32, 41]. Heterogeneity may impact antibiotic and clearance efficacy if portions of the biofilm are more recalcitrant to chemical or mechanical challenges, but the translational outcomes of biofilm heterogeneity have not yet been investigated. Further, increasing heterogeneity may be indicative of greater biofilm disruption. Indeed, every treatment was associated

with an increase in κ and thus heterogeneity with the greatest increase being observed in TCEP and TCEP+Tob treated biofilms, 55.75 and 28.08 respectively. The changes in complex viscosity distribution after treatment can be visualized in S2 Fig in S1 File, which highlights the changing biophysical properties of the biofilms with treatment.

## Mucociliary transport of treated *P. aeruginosa* biofilms is heterogeneous

The presence of planktonic bacteria in 2% or 3% mucus was associated with significantly reduced MCT rates despite nominally altered η* (S3 Fig in S1 File). Previously, bacterial products have been shown to decrease CBF which may reduce MCC [42]. These data indicate that the addition of planktonic bacteria to mucus compromises MCT rates. Biofilm MCT rates were markedly decreased compared to uninfected mucus (Fig 5) despite having relatively similar η* values. For treated biofilms, there was no observable trend as a function of matrix disruption (NP40, TCEP, HA, or DNase) or the combination of Tob (Fig 5). The greatest mean MCT rate was observed in NP40 treated biofilms at 2.9 μm/s. Statistically significant increases in MCT rates were achieved with NP40 and DNase+Tob treatments. The spatial heterogeneity of particle transport in TCEP+Tob treated biofilms is represented in S4 Fig in S1 File. Transport of single and dual agent treated biofilms shows that NP40 was the most effective at increasing mean MCT rates, but DNase and Tob combinations were superior to other

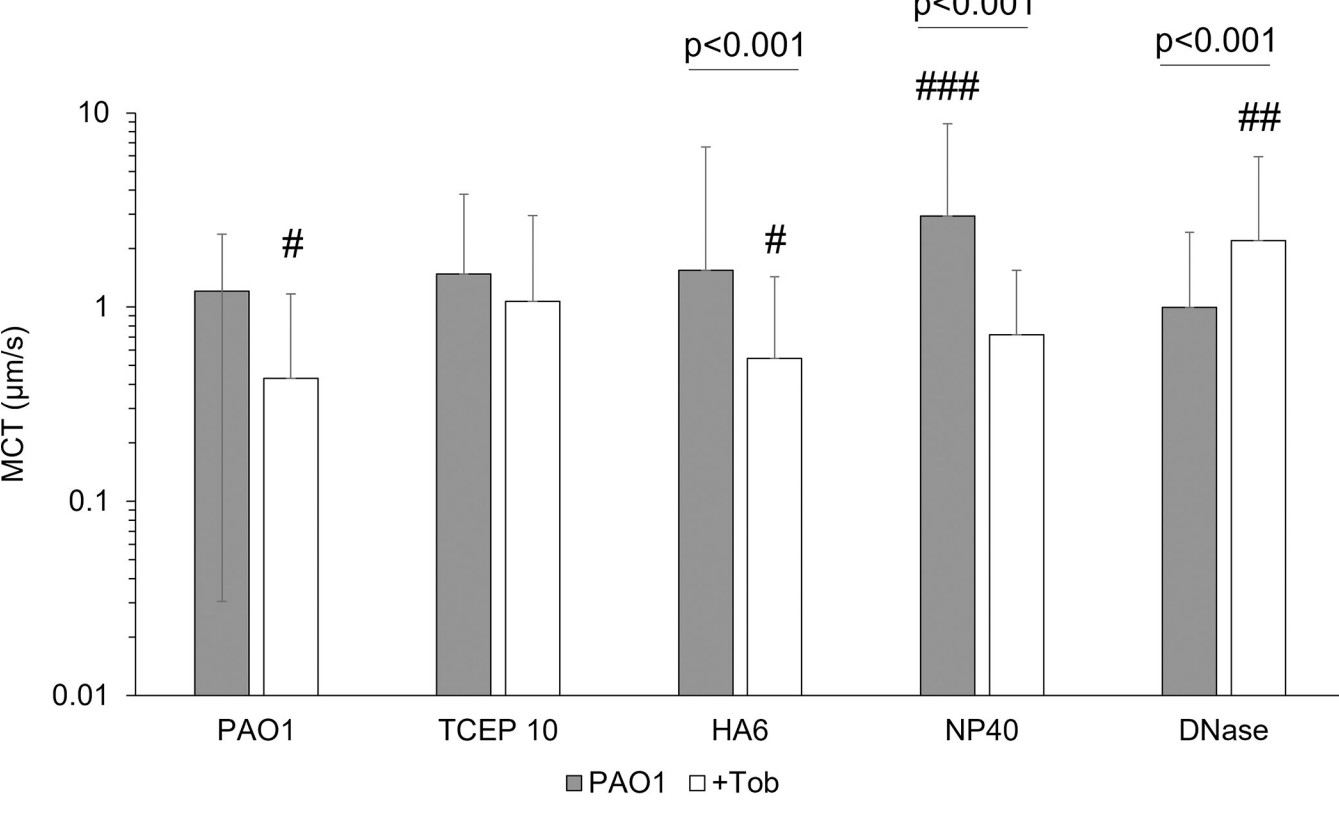

# p<0.05, ## p<0.01, ### p<0.001 compared to PAO1

**Fig 5. Mucociliary transport of biofilms in the racetrack culture.** Single agent treatment is shown in gray and combination with Tob is shown in white. Data is representative of the mean ± standard deviation of every tracked bead with a MCT rate >0.1 μm/s from ≥6 videos from separately prepared and evaluated biofilms. Statistical significance determined via single factor ANOVA with post hoc Tukey Kramer analysis.

combinations with Tob. Tob treatment alone decreased biofilm MCT. It is possible that cell death due to antibiotic treatment resulted in sufficiently altered biophysical properties of the biofilm to result in decreased MCT rates, which was observed for HA+Tob and NP40+Tob, despite the increased in MCT observed with single agent treatment. These data indicate that dual combination treatment is insufficient to optimize biofilm disruption and clearance.

### Multimechanistic biofilm disruption results in increased mucociliary transport rates

Combinations of treatments were prepared to evaluate biofilm viscoelasticity and transport after disruption of the biofilm matrix using multiple mechanisms: 1) HA+NP40+Tob, 2) TCEP+NP40+Tob, and 3) TCEP+HA+Tob. These combinations were selected to optimize a) reductions in biofilm $\eta^*$, b) increases in MCT rates, and c) bactericidal action (Fig 6). Both combinations that included TCEP resulted in increased mean $\eta^*$ values compared to Tob treatment alone or the untreated control (Fig 6A). However, the HA+NP+Tob combination resulted in a decreased mean $\eta^*$ with a broad range of measured $\eta^*$ by individual particles. Uniquely, this combination also caused a shift in viscoelastic behavior of the biofilm from being dominated by the elastic modulus (G'), as seen in the untreated control (Fig 6B) to being dominated by the viscous modulus (G"). While the untreated biofilm exhibits viscoelastic solid behavior (G'>G"), the NP40+HA+Tob treatment results in a transition to viscoelastic liquid behavior (G">G'), which is indicative of breakdown of the polymeric mesh. Further, HA+NP40+Tob treatment was consistently capable of eradicating bacterial viability to below the limit of detection for colony counting ($10^3$ CFU/mL, Fig 6C). All three preparations of triple combination treatment resulted in significantly (p<0.001) greater MCT rates compared to the untreated biofilm (Fig 6D). Finally, macrorheology of triple combination treated biofilms determined that HA+NP+Tob restored macroscopic $\eta^*$ to that of uninfected 3% HBE (S5 Fig in S1 File).

## Discussion

There is increasing evidence that bacterial aggregation and biofilm formation are responsible for both acute and chronic infections [11, 12], which is likely due in part to increased protection from mechanical stressors like mucociliary clearance and phagocytosis as well as chemical challenges from the immune response and antibiotics. *Pseudomonas aeruginosa* biofilms grown in HBE mucus exhibit significantly different rheological properties compared to uninfected mucus and mucus with planktonic bacteria (Fig 4B). The biophysical properties of *P. aeruginosa* biofilms are influenced by mucus concentration [25, 32]. Mucus similarly exhibits increasing $\eta^*$ values with concentration (Fig 3A), and pathological, hyperconcentrated mucus is associated with decreasing lung function over time [4, 5, 10]. The racetrack MCTD represents an ideal screening tool for evaluating therapeutic potential of antibacterial compounds designed to disrupt mucus or biofilms and increase the capacity for clearance from the airway. The work described herein identifies a triple combination treatment capable of 1) reducing bacterial viability, 2) altering biofilm viscoelastic properties, and 3) increasing mucociliary transport rates.

Cilia beat frequency and MCT rates of beads in PBS were consistent between quadrants of an individual racetrack and across separate racetrack cultures. Comparison of PTMR data (Fig 3A) and MCT rates (Fig 3C) of HBE mucus indicated a power law relationship between $\eta^*$ and MCT rates (Fig 3D). Notably, 2% mucus $\eta^*$ and 5%+TCEP $\eta^*$ values were similar (0.03 Pa·s and 0.04 Pa·s, respectively) and MCT rates were comparable (~8±2 μm/s and 3±3 μm/s, respectively). Thus, the measured $\eta^*$ of mucus may be used to approximate MCT rates.

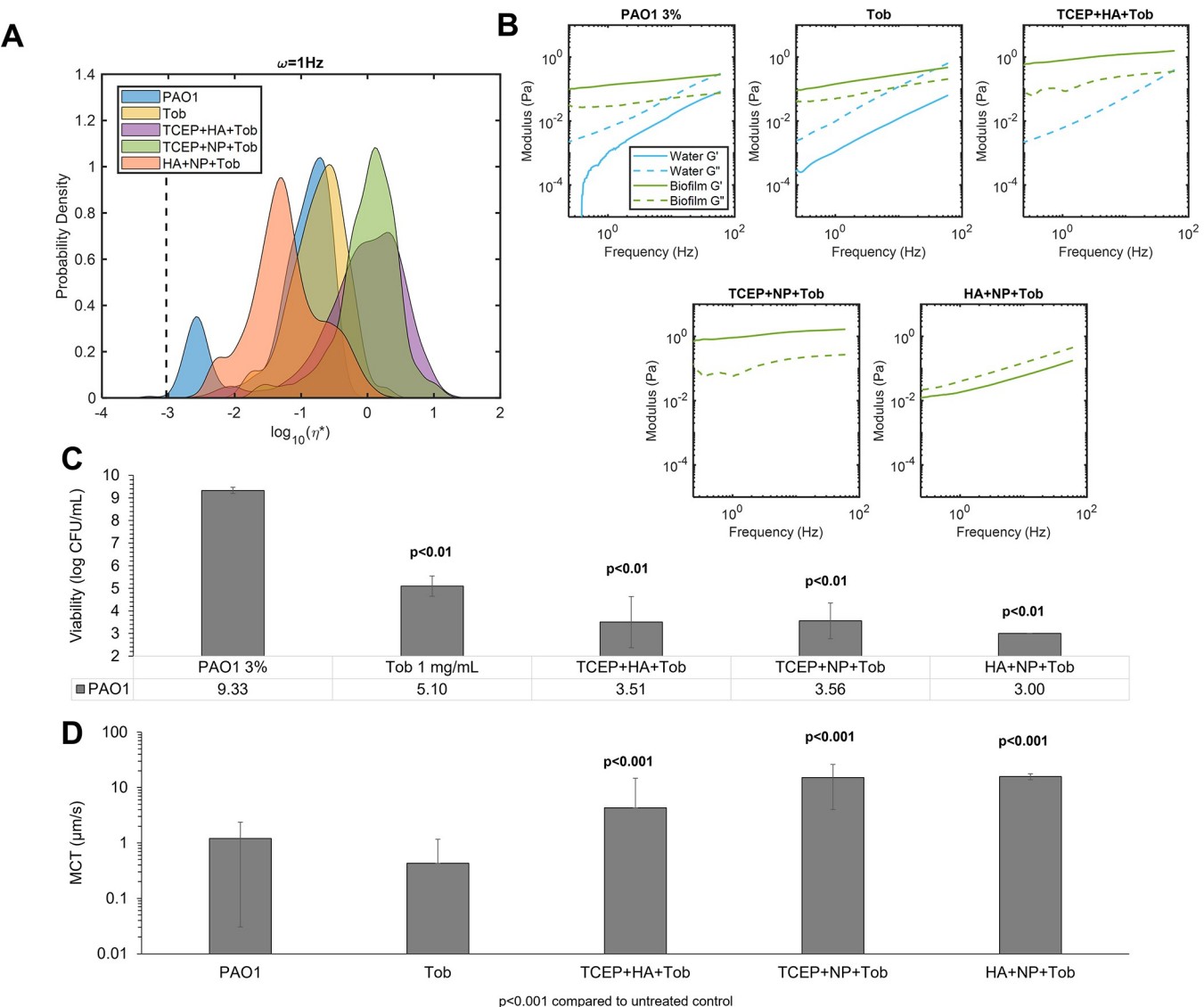

**Fig 6. Triple combination treatment of PAO1 biofilms.** A) Biofilm complex viscosity distribution as a function of treatment conditions. Data is representative of every tracked particle complex viscosity for three separately prepared and evaluated specimens. B) Biofilm microscopic moduli as a function of treatment. The elastic modulus (G', solid) and the viscous modulus (G", dashed) are shown in green for the solid-like biofilm component and in blue for the more watery component as determined via Gaussian mixture modeling. C) Viability of PAO1 biofilms as a function of triple combination treatment. Significance was determined using single factor ANOVA with post hoc Tukey analysis. D) Mean MCT rates of biofilms as a function of triple combination treatment. Data is presented as the mean ± standard deviation of tracked particles for n≥6 videos. Significance was determined using single factor ANOVA with post hoc Tukey Kramer analysis.

Surprisingly, this relationship does not apply to *P. aeruginosa* biofilms grown in HBE mucus. There was no observable trend in biofilm η* (Fig 3A) on MCT (Fig 5) in the racetrack. In fact, the simple addition of planktonic bacteria to 2% or 3% mucus was associated with significantly reduced transport despite nominal changes in η* (S3 Fig in S1 File). This phenomenon may be due in part to altered CBF due to bacterial products, which has previously been shown [42]. Indeed, CBF decreased by nearly half 24 h after initial measurements of biofilm MCT in the racetrack (S6 Fig in S1 File). Thus, treatments must sufficiently disrupt the biofilm architecture and eradicate bacteria to prevent decreases in CBF. Combination treatments with

**Table 2. The non-Gaussian parameter κ as a function of treatment conditions for n≥4 separately prepared and evaluated biofilms.** Larger values are associated with increasing heterogeneity.

| Treatment | κ |
|---|---|
| PAO1 3% | 0.09 |
| Tob | 2.03 |
| NP40 | 4.82 |
| NP40+Tob | 7.60 |
| TCEP | 55.75 |
| TCEP+Tob | 28.08 |
| HA | 4.65 |
| HA+Tob | 14.01 |
| DNase | 4.50 |
| DNase+Tob | 3.00 |

tobramycin resulted in significantly decreased bacterial viability (Fig 4C) but minimal impact on biofilm $\eta^*$. HA treatment alone significantly increased biofilm $\eta^*$ and has previously been shown to potentiate the antibiotic efficacy of Tob [25]. Indeed, none of the disruption agents in combination with tobramycin resulted in antagonistic bactericidal action (Fig 4C). However, the lack of significant alterations of biofilm $\eta^*$ and broad heterogeneity in MCT rates (Fig 5) motivated the investigation of triple combination treatment to maximize biofilm disruption and clearance.

Multi-mechanistic targeting of bacterial infections is an attractive pathway for therapeutic development due to the potential for synergistic interactions, mitigating antibiotic resistance development, and improved patient compliance [43–45]. The most effective biofilm disruption agents characterized with PTMR (NP40 and TCEP) were combined with the most effective transporting treatments (TCEP and HA) and Tob to produce a triple combination for optimizing changes in biofilm $\eta^*$ and increases in MCT (Fig 6). Further, the greatest changes in κ values (Table 2) associated with combinations with tobramycin included TCEP, NP40, and HA. Thus, triple combination treatments were prepared with these three agents and tobramycin (see Table 1). The most effective triple combination of HA+NP+Tob caused a shift from elastic-dominated to viscous dominated behavior of the biofilm, indicating a severely compromised polymeric mesh (Fig 6B) and completely reduced bacterial viability to below the limit of detection (Fig 6C). Further, MCT rates were significantly increased after treatment with HA +NP+Tob (Fig 6D). This triple combination may be efficacious in treating chronic biofilm respiratory infections in MOPD such as CF.

## Conclusion

Respiratory infections in MOPD have consistently been difficult to eradicate permanently due in large part to the viscoelastic properties of pathological mucus that are associated with mechanically robust biofilms and antibiotic resistance [1, 20]. Mucus $\eta^*$ is correlated to MCT rates in the racetrack MCTD via a power-law relationship where more viscoelastic mucus exhibits worse transport. *Pseudomonas aeruginosa* biofilms, however, exhibit no such relationship which may be due in part to interactions between bacterial cells and epithelial cilia. The presence of planktonic bacteria in mucus significantly reduced MCT rates despite having little impact on mucus $\eta^*$. Biofilm disruption, characterized as either an increase or decrease in $\eta^*$, and increased MCT were optimized through combination treatments of a surfactant NP40, the reducing agent TCEP, a biopolymer HA, and the antibiotic Tob. The most effective triple combination of HA+ NP40+Tob significantly compromised the biofilm architecture, resulting

in a transition from elastic-dominated to viscous-dominated behavior, and reduced bacterial viability to below the limit of detection. Further, biofilm MCT rates were markedly increased after treatment with HA+ NP40+Tob. This triple combination will be prepared for tolerability studies and antibacterial efficacy evaluations in an infected muco-obstructive murine model in future work.

## Supporting information

**S1 File. This file contains supporting table and supporting figures.**
(DOCX)

## Author Contributions

**Conceptualization:** Kaitlyn R. Rouillard.

**Data curation:** Kaitlyn R. Rouillard, Matthew R. Markovetz.

**Formal analysis:** Kaitlyn R. Rouillard, Lucas M. Plott, Matthew R. Markovetz, David B. Hill.

**Funding acquisition:** Kaitlyn R. Rouillard, David B. Hill.

**Investigation:** Kaitlyn R. Rouillard, Christopher P. Esther, Lucas M. Plott, Dean W. Bowman.

**Methodology:** Kaitlyn R. Rouillard, William J. Kissner, Matthew R. Markovetz, David B. Hill.

**Project administration:** Kaitlyn R. Rouillard, David B. Hill.

**Resources:** William J. Kissner, Lucas M. Plott, David B. Hill.

**Supervision:** Kaitlyn R. Rouillard, David B. Hill.

**Validation:** Kaitlyn R. Rouillard, Matthew R. Markovetz.

**Writing – original draft:** Kaitlyn R. Rouillard.

**Writing – review & editing:** Kaitlyn R. Rouillard, David B. Hill.

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
