## [Decision Letter · Decision Letter 0]

10 Oct 2023

PONE-D-23-29054Combination Treatment to Improve Mucociliary Transport of Pseudomonas aeruginosa BiofilmsPLOS ONE

Dear Dr. Hill,

Thank you for submitting your manuscript to PLOS ONE. After careful consideration, we feel that it has merit but does not fully meet PLOS ONE’s publication criteria as it currently stands. Therefore, we invite you to submit a revised version of the manuscript that addresses the points raised during the review process.

We look forward to receiving your revised manuscript.

Kind regards,

Abdelwahab Omri, Pharm B, Ph.D, Laurentian University

Academic Editor

PLOS ONE

Journal Requirements:

3. Thank you for stating the following financial disclosure: "KRR (Cystic Fibrosis Foundation ROUILL22F0)

DBH (Cystic Fibrosis Foundation HILL19G0, HILL20y2-OUT) 

This work also benfited from Cores funded by the Cystic Fibrosis Foundation (BOUCHE19R0) and NIH (1P01HL164320)".

4. Thank you for stating the following in the Acknowledgments Section of your manuscript: "Funding for this work was provided by the Cystic Fibrosis Foundation (ROUILL22F0, HILL19G0), HILL20Y2-OUT, BOUCHE19R0), and the NIH (P30DK065988 and 1P01HL164320)".

Please remove any funding-related text from the manuscript and let us know how you would like to update your Funding Statement. Currently, your Funding Statement reads as follows: "KRR (Cystic Fibrosis Foundation ROUILL22F0)

DBH (Cystic Fibrosis Foundation HILL19G0, HILL20y2-OUT) 

This work also benfited from Cores funded by the Cystic Fibrosis Foundation (BOUCHE19R0) and NIH (1P01HL164320)"

5. Thank you for stating the following in your Competing Interests section: "No authors have competing interest".

6. We note that you have indicated that data from this study are available upon request. PLOS only allows data to be available upon request if there are legal or ethical restrictions on sharing data publicly. For more information on unacceptable data access restrictions, please see http://journals.plos.org/plosone/s/data-availability#loc-unacceptable-data-access-restrictions. 

7. Please note that in order to use the direct billing option the corresponding author must be affiliated with the chosen institute. Please either amend your manuscript to change the affiliation or corresponding author, or email us at plosone@plos.org with a request to remove this option.

8. Please include your full ethics statement in the ‘Methods’ section of your manuscript file. In your statement, please include the full name of the IRB or ethics committee who approved or waived your study, as well as whether or not you obtained informed written or verbal consent. If consent was waived for your study, please include this information in your statement as well.

9. Please upload a copy of Supporting Information Figure/Table/etc. Supplemental Figure 1, 2, 3, 4, 5 and 6 which you refer to in your text on pages 21, 22, 23 and 24.

Reviewers' comments:

Reviewer's Responses to Questions

**Comments to the Author**

1. Is the manuscript technically sound, and do the data support the conclusions?

Reviewer #1: Yes

Reviewer #2: Partly

2. Has the statistical analysis been performed appropriately and rigorously? 

Reviewer #1: Yes

Reviewer #2: Yes

3. Have the authors made all data underlying the findings in their manuscript fully available?

Reviewer #1: Yes

Reviewer #2: Yes

4. Is the manuscript presented in an intelligible fashion and written in standard English?

Reviewer #1: Yes

Reviewer #2: Yes

5. Review Comments to the Author

Reviewer #1: The manuscript reports a study on combinations of antibiotic with biofilm matrix or/and mucus modulating actives to improve chronic bacterial infections therapy in patients with muco-obstructive diseases. The aim of the drug combinations is to simultaneously kill the bacterial pathogen and increase the mucociliary clearance. The effect of mono and dual and triple drug combinations was investigated i) on rheology of mucus biofilm ii) ciliary transport rates using an in vitro cell system and iii) bacteriocidal efficacy against P. aeruginosa (PAO1).

The study is well planned, performed and described. A model to investigate the mucociliary clearance in controlled in vitro setting without ethical concerns of animal experiments is an interesting set-up which could help in development of mucolytic formulation of anti-infectives. The selected drug combinations are based on authors experience and literatur and are well chosen.

Few minor adjustments are recommended before acceptance in PLOS ONE for publishing.

• Please spell out the full name, or even further specification of the surfactant NP40 (e.g., avoiding possible confusion with Nonidet-P40).

• NP40 can be used for cell lysis. Some comment on the used concentration of this surfactant would be welcome

• In MCT rates determination, the procedure is described to filter out stuck beads (i.e. MC<01. µm/s). Figure 2 reports CBFs and MCTs are sufficiently reproducible over cultures and quadrants. Is the removal of stuck beads essential to achieve the reproducibility or only facilitating the calculation? Is the fraction of stuck beads approx. equal for quadrants and culture samples?

• Supplemental Figure 4: Please clarify in the caption if the 4 Videos show 4 technical replicates or different regions form the same sample (treatment 10mM TCEP+ 1mg/mL Tob).

Reviewer #2: The data presented here make a compelling case for combining therapies targeting mucus clearance with antibiotics in order to more effectively eradicate chronic Pseudomonas aeruginosa infections. The triple combination approach is interesting, and the data showing that viscoelastic behavior of the biofilm switching from a viscoelastic solid to a viscoelastic liquid is particularly compelling. However, there are some weaknesses that the authors should consider to enhance the impact of the data.

1. The experiments are done only on non-diseased human bronchial cells. Adding in cells from a donor with muco-obstructive pulmonary disease would be important to understand if this approach will be efficacious in disease.

2. Table 1 is helpful for the reader but not entirely clear. Why was the listed combination chosen? In particular, why was DNAse not included in a triple combination approach given that it is a currently approved therapeutic?

3. In figure 4B, it is not clear which data set indicates biofilm vs planktonic forms of the bacteria.

4. In Figure 4D, the change in CFUs is important, but it may be more impactful to compare the CFUs to tobramycin, since this is the current standard of care.

5. Table 2, and the biological impact of this data, is not clear.

6. In Figure 5, it is not clear why the MCT rates go down with the addition of tobramycin for all the groups except for DNAse. Are there CFU data to match this experiment?

7. It is understandable why the authors chose to start the experiments with PAO1. It will be much more impactful if these results are confirmed in the presence of a clinical isolate or mucoid strain.

6. PLOS authors have the option to publish the peer review history of their article (what does this mean?). If published, this will include your full peer review and any attached files.

Reviewer #1: No

Reviewer #2: No

---

## [Author Response · Author response to Decision Letter 0]

20 Oct 2023

We thank the reviewers for their kind and constructive review of our manuscript. We have addressed each concern with tracked changes in the submitted document. Below are our point-by-point replies:

Reviewer 1:

1. NP40 is nonyl phenoxypolyethoxylethanol. We have updated the manuscript to include this full name.

2. NP40 was used at 0.1% which is below the critical micelle concentration. We wanted to avoid any complications from micelle formations in the rheological impact of treatment. Additionally, the 0.1% dose was the lowest dose necessary to induce significant decreases in macroscopic and microscopic complex viscosity values in both 2% and 4% mucus. These data are included in a different manuscript submission specific to mucus rheology. We have referenced the previous study that included the dosages used for the other components. 

3. The removal of the stuck beads is critical for an accurate quantification of MCT rates. A large portion of the beads have effectively transport rates of zero, which decreases the ensemble rate greatly. The number of stuck beads across videos ranges from 0-25, depending on the properties of the transporting sample. More heterogeneous and viscous samples are associated with more stuck beads, such as untreated biofilms and mucus + planktonic bacteria. The treated biofilms that have a compromised matrix, particularly those treated with the triple combinations, have far fewer stuck beads. For transparency, we have included the number of stuck beads removed from each sample into a table in the supporting information. 

4. Each video is acquired in a different region of the culture. We have updated the caption to specify this. 

Reviewer 2:

1. Transport in diseased HBE cells can be especially challenging, particularly with decreased MCT rates in the diseased cultures. Also, with the decreasing availability of CF lungs, CF cells are being used for more precious experiments. We are currently investigating the use of conditionally reprogrammed cells (CRC) to expand CF cells to compare mucus biophysical properties. When these studies are complete, we will be able to utilize CF HBE mucus and cells for transport experiments. For the purpose of this study, HBE mucus was prepared at 3% solids to represent mild pulmonary disease. We believe that the present work lays the foundation necessary for more complex studies with diseased HBE cells, like CF HBE CRC cultures, that can then benefit from a dose-response study with the triple combination treatments screened in this study. 

2. Treatments that maximized biofilm disruption (increase or decrease in biofilm complex viscosity) and reductions in bacterial viability were combined with tobramycin for the triple combination treatment. DNase was excluded from triple combination treatments due to minimal impact on biofilm rheology in high mucus concentrations, which we have published on previously. TCEP, NP40, and HA have demonstrated more promising biofilm disruption capabilities in this study and our previous work. This study laid the groundwork necessary to evaluate a single treatment (TCEP/NP40/HA) in an infected murine model, which is upcoming. 

3. Throughout the document, PAO1 3% is used to indicate biofilms grown in 3% mucus while 3% HBE + PAO1 is used to indicate mucus with planktonic bacteria. The caption includes this information as well. We have also included a clarifying statement into the text to highlight the nomenclature and remove any confusion. 

4. The individual agents do not significantly decrease bacterial viability, and ultimately, the goal is to optimize reductions in biofilm complex viscosity and viability. Thus, we must ensure that there are no antagonistic interactions between the biofilm disruption agent (e.g., TCEP) and the antibiotic. 

5. We have previously published on the non-Gaussian parameter (κ) for PAO1 biofilms grown in mucus. Greater κ values indicate greater heterogeneity of the biofilm biophysical properties, which may be used as a measure of biofilm disruption. The combinations that induced the greatest changes in κ values were TCEP, HA, and NP40, each with tobramycin. The PTMR data for each of these conditions is included in Supplemental Figure 2 where we qualitatively observe changes in biofilm biophysical behavior (i.e., complex viscosity distribution). These values suggest that combinations of these agents may result in superior biofilm disruption that may be associated with increased MCT rates. Indeed, the triple combinations of these agents significantly increased MCT rates. We have added a statement to this effect into the discussion section. 

6. The CFU data for the combinations are included in Figure 4, but treatment with the single agent had little effect on viability. We hypothesize that biofilm disruption with the single agent (e.g., TCEP, HA) was sufficient to increase MCT rates but not eradicate bacteria. However, the addition of tobramycin resulted in cell death that contributed to slightly increased complex viscosity values (4C) and decreased MCT rates (Fig 5). We have included a statement to this effect in the results section. 

7. We have a reproducible and robust biofilm growth model in HBE mucus with PAO1 for in vitro studies. The benefit to PAO1 for these experiments is its broad availability to other labs as well. The goal of this work was to screen and evaluate combination treatments in order to select an optimal in vitro candidate for in vivo studies. Because the in vitro susceptibility of even clinical isolates is often different from in vivo behavior, we wanted to optimize biophysical changes with PAO1 before moving to an isolate that’s been optimized for an infected murine model. These studies are ongoing.

---

## [Editor Report · Decision Letter 1]

26 Oct 2023

Combination Treatment to Improve Mucociliary Transport of Pseudomonas aeruginosa Biofilms

PONE-D-23-29054R1

Dear Dr. David B Hill,

We’re pleased to inform you that your manuscript has been judged scientifically suitable for publication and will be formally accepted for publication once it meets all outstanding technical requirements.

Kind regards,

Abdelwahab Omri, Pharm B, Ph.D, Laurentian University

Academic Editor

PLOS ONE

---

## [Editor Report · Acceptance letter]

10 Nov 2023

PONE-D-23-29054R1 

Combination Treatment to Improve Mucociliary Transport of Pseudomonas aeruginosa Biofilms 

Dear Dr. Hill:

I'm pleased to inform you that your manuscript has been deemed suitable for publication in PLOS ONE. Congratulations! Your manuscript is now with our production department. 

Kind regards, 

on behalf of

Dr. Abdelwahab Omri 

Academic Editor

PLOS ONE